# GnomAD Missense Variants of Uncertain Significance: Implications for p53 Stability and Phosphorylation

**DOI:** 10.3390/ijms26157455

**Published:** 2025-08-01

**Authors:** Fernando Daniel García-Ayala, María de la Luz Ayala-Madrigal, Jorge Peregrina-Sandoval, José Miguel Moreno-Ortiz, Anahí González-Mercado, Melva Gutiérrez-Angulo

**Affiliations:** 1Instituto de Genética Humana “Dr. Enrique Corona Rivera”, Centro Universitario de Ciencias de la Salud, Universidad de Guadalajara, Guadalajara 44340, Jalisco, Mexico; fernando.garcia9652@alumnos.udg.mx (F.D.G.-A.); luz.ayala@academicos.udg.mx (M.d.l.L.A.-M.); miguel.moreno@academicos.udg.mx (J.M.M.-O.); anahi.gonzalez@academicos.udg.mx (A.G.-M.); 2Programa de Doctorado en Genética Humana, Centro Universitario de Ciencias de la Salud, Universidad de Guadalajara, Guadalajara 44340, Jalisco, Mexico; jorge.peregrina@academicos.udg.mx; 3Departamento de Biología Celular y Molecular, Centro Universitario de Ciencias Biológicas y Agropecuarias, Universidad de Guadalajara, Zapopan 44600, Jalisco, Mexico; 4Departamento de Ciencias de la Salud, Centro Universitario de los Altos, Universidad de Guadalajara, Tepatitlán de Morelos 47600, Jalisco, Mexico

**Keywords:** *TP53* gene, VUS, phosphorylation sites, protein stability, gnomAD

## Abstract

The *TP53* gene, frequently mutated across multiple cancer types, plays a pivotal role in regulating the cell cycle and apoptosis through its protein, p53. Missense variants of uncertain significance (VUSs) in *TP53* present challenges in understanding their impact on protein function and complicate clinical interpretation. This study aims to analyze the effects of missense VUSs in p53, as reported in the gnomAD database, with a specific focus on their impact on protein stability and phosphorylation. In this study, 33 missense VUSs in *TP53* reported in the gnomAD database were analyzed using in silico tools, including PhosphositePlus v6.7.4, the Kinase Library v0.0.11, and Dynamut2. Of these analyzed variants, five disrupted known phosphorylation sites, while another five created new consensus sequences for phosphorylation. Moreover, 20 variants exhibited a moderate destabilizing effect on the protein structure. At least three missense VUSs were identified as potentially affecting p53 function, which may contribute to cancer development. These findings highlight the importance of integrating in silico structural and functional analysis to assess the pathogenic potential of missense VUSs.

## 1. Introduction

Reference population databases are essential for understanding genetic variation and identifying rare variants that may contribute to disorders. The Genome Aggregation Database (gnomAD), originally launched in 2014 as the Exome Aggregation Consortium (ExAC), is the largest publicly available resource for human genome allele frequencies. It was created by aggregating sequencing data from multiple large-scale projects, such as the Human Genome Diversity Project and the 1000 Genomes Project, processing all data through standardized pipelines and jointly calling variants to ensure consistency. Version 3.1.2 of gnomAD includes 730,947 exome sequences and 76,215 whole-genome sequences from unrelated individuals of diverse ancestries and reports 644,267,978 variants [1]. Variants can be filtered based on their impact, and specialized databases should also be consulted for annotations related to pathogenicity and functional consequences. Some variants, however, do not meet classification criteria and are categorized as variants of uncertain significance (VUSs). Among these, missense VUSs involve amino acid substitutions that may affect protein structure, stability, or function. Despite this potential, the precise impact of the substitutions on protein function and their role in disease development remain unclear [2], leaving uncertainty about whether they contribute to disease susceptibility or are benign variations [3]. This ambiguity is particularly significant for genes essential for cellular functions, such as *TP53*.

The *TP53* gene, located at 17p13.1, consists of 11 exons and encodes the cellular tumor antigen p53 protein, a 393-residue protein with a molecular weight of 43,653 Daltons (Uniprot ID-P04637) [4]. p53 has intrinsically disordered N- and C-terminal regions and a highly structured central region that contains a DNA-binding domain (DBD) [5,6,7,8]. The N-terminal region contains transactivation domains (TAD1 residues 6–30 and TAD2 residues 35–59) that enhance p53’s ability to locate high-affinity DNA binding sites. Additionally, it interacts with transcription machinery, including TBP-associated factors such as TFIIH, as well as chromatin-modifying proteins like CBP/p300 and GCN5 [6]. The DBD (residues 100–288) relies on a Zn^2+^ ion for structural integrity and DNA-binding specificity [9]. It has eight essential residues (K120, R241, K248, K273, A276, A277, R280, and R283) that establish direct hydrogen bonds with DNA, enabling the DBD to bind specific target sequences and position the tetramer on the genes it regulates [10]. The C-terminal region, which is flexible and highly basic, serves as a binding site for regulatory proteins and interacts with DNA to facilitate p53’s movement and identifies recognition elements [6]. This region also contains the tetramerization domain (TET, residues 319–357), essential for the quaternary structure of p53 and its function [11].

As a multifunctional nuclear transcription factor, p53 regulates genes involved in DNA damage response, apoptosis, and cell cycle arrest and senescence, playing a critical role in cellular homeostasis and tumor suppression [12]. Under normal conditions, p53 levels are kept low through a negative feedback loop, where p53 induces the expression of MDM2. The MDM2 protein binds to the N-terminal region of p53 and acts as an E3 ubiquitin ligase, tagging p53 for nuclear export and proteasomal degradation [13]. Upon stress signals such as DNA damage, oncogene activation, or hypoxia, p53 ubiquitination is interrupted via post-translational modifications (PTMs), with phosphorylation being one of the most significant. Kinases such as ATM, ATR, and CHK1/CHK2 phosphorylate p53 at specific sites, promoting its activation and preventing degradation by MDM2. Phosphorylation also facilitates interactions with specialized reader proteins, such as the 14-3-3 protein, which modulates p53’s stability and transcriptional activity, working in concert with other PTMs like acetylation and ubiquitination [14]. As a result, p53 accumulates in the nucleus as an active tetramer, binds to specific DNA sequences, and regulates the transcription of numerous target genes involved in cell cycle arrest, allowing the cell time and resources to repair genomic damage. [15]. If repair fails, p53 switches to an apoptotic program and upregulates pro-apoptotic *BCL2* family genes such as *NOXA, PUMA, BID, BAD, BIK,* and *BAX* [16].

The *TP53* gene presents variants in approximately 50–60% of human cancers. Most of these alterations (~90%) correspond to missense variants concentrated in the central region of p53 [17]. According to the *TP53* Database (version R21, ISB-CGC, https://TP53.isb-cgc.org/, accessed on 27 July 2025), hotspot residues R175, R213, G245, R248, R249, R273, R282, and R337 account for approximately 25% of all *TP53* mutations and hold significant clinical relevance. These alterations have been detected in both somatic and germline contexts, which affect the protein structure, reduce its DNA-binding capacity, and disrupt the transcriptional regulation of key genes involved in the cell cycle and tumor suppression [18,19,20].

When *TP53* missense variants are classified as VUSs either in the somatic or germline context and are in functional regions potentially affecting protein structure, stability, DNA binding, protein–protein interactions, or consensus sites for PTM [21], they represent a significant challenge for interpretation. Despite the application of ACMG/AMP guidelines and *TP53*-specific recommendations, a substantial proportion of these variants remain unresolved [22].

In this study, we analyzed VUSs reported in gnomAD v3.1.2 (https://gnomad.broadinstitute.org/, accessed on 4 June 2024), assessing their impact on p53 protein stability and phosphorylation sites using bioinformatics predictors.

## 2. Results

The analysis of *TP53* in gnomAD v.3.1.2. revealed 186 missense variants, which were further examined in free VarSome (https://varsome.com/, accessed on 4 June 2024) to select only missense VUSs. This search resulted in 33 variants. The distribution of exons and the frequency of the minor allele of each variant are detailed in Table 1. Of the 33 variants, 27.3% (9/33) were in the N-terminal region, 42.4% (14/33) in the central region, and 30.3% (10/33) in the C-terminal region. Notably, in the N-terminal within the TAD1 domain, two variants were located at residue 11 (p.E11K and p.E11Q), while two variants were within central region, specifically in the DBD at residue 129 (p.A129T and p.A129S) (Table 1). Figure 1 shows the distribution of missense VUSs across p53, along with the proteins that interact with the affected residues.

### 2.1. Impact of VUS Missense on Phosphorylation Sites

The PhosphoSitePlus v6.7.4 database (https://www.phosphosite.org/homeAction.action, accessed on 4 June 2024) described a total of 52 PTM-susceptible sites in p53, including phosphorylation, acetylation, ubiquitination, methylation, and SUMOylation. Specifically, regarding phosphorylation, 15% of the missense VUSs (5 out of 33), resulted in the loss of canonical phosphorylation residues. Moreover, in the kinase library v0.0.11 database, 5 of the 33 variants were found to introduce a new PhS recognized by different kinases, as shown in Figure 2. The results are summarized in Table 1.

### 2.2. Stability Analysis of Missense VUSs

To assess their impact on protein stability, missense VUSs were analyzed using the Dynamut2 platform (https://biosig.lab.uq.edu.au/dynamut2/, accessed on 4 June 2024). The results indicated that 61% (20/33) were moderately destabilizing (Table 1). The N-terminal region containing TAD1 and TAD2 included nine variants affecting eight residues. Notably, only p.P13L, located in TAD1, exhibited destabilizing effects, as illustrated in Figure 3a. In the central region, a total of 13 residues affected by 14 variants were analyzed. The evaluated missense VUSs allowed us to identify 12 moderately destabilizing variants (86%) (Figure 3b). Finally, ten residues affected in the C-terminal region, which includes TET, were analyzed. Stability analysis revealed that seven (70%) of these variants had a moderately destabilizing effect, as detailed in Figure 3c.

### 2.3. Conservation Analysis of Missense VUSs

An evolutionary conservation analysis was conducted using the PhyloP metric, comparing 100 vertebrate species with the human reference genome (GRCh38/hg38 assembly). The PhyloP metric quantifies the deviation of nucleotide evolution from neutrality at each genomic position. Among the 33 values, 76% (25/33) were positive (Table 1), indicating the presence of evolutionarily conserved regions across vertebrate species and the possible preservation of protein structure and function. Moreover, a comprehensive analysis was conducted for the T155 residue, which demonstrated that threonine and serine occur in 58% and 23% of species, respectively, while alanine and cysteine are present in 18% and 1%, respectively.

## 3. Discussion

In the last decade, advancements in genetic testing, driven by high-throughput technologies for genome and exome analysis, have led to the creation of databases like gnomAD, which aggregates data from diverse populations [1]. The *TP53* data in gnomAD are crucial for understanding the prevalence of germline variants, many of which are rare and of uncertain clinical significance, leaving their role in cancer unclear [23]. Additionally, the impact of missense variants can vary based on their location in relation to the protein, potentially affecting its function and/or stability, which presents a challenge for clinical genomics and variant interpretation [24,25]. This study focused on the impact of 33 missense VUSs on the phosphorylation and stability of the p53 protein. Of these, 10 were related with alterations in PhS and 20 variants resulted in protein structure destabilization. Five variants affected both PhS and structural stability.

Protein phosphorylation plays a critical role in various cellular processes by altering its three-dimensional structure and/or modifying the protein–protein interactions. The p53 protein is crucial for numerous cellular processes, and its levels, activity, and localization are tightly regulated by PTM including ubiquitination, acetylation, SUMOylation, and phosphorylation [26]. In this study, in silico analysis revealed that the missense VUSs p.S9N, p.S15N, p.T155S, p.S362I, and p.S378C lead to the loss of residues susceptible to phosphorylation, while p.L114S, p.C124S, p.A129T, p.A129S, and p.T155S generated new potential PhS.

The variants p.S9N, p.S15N, p.T155S, p.S362I, and p.S378C involve the substitution of serine or threonine residues, which may disrupt key phosphorylation sites, potentially impacting p53 function.

The in silico analysis revealed that p.S9N and p.S15N result in the loss of phosphorylation sites. The p.S9N variant is described in Li–Fraumeni syndrome in the ClinVar database, and it has been reported in a melanoma patient in the cBioPortal database, while p.S15N lacks any reports in the ClinVar, cBioPortal, and COSMIC databases [27,28,29]. Serine 15 is phosphorylated by ATM and ATR, which enables subsequent phosphorylation at S9, S20, and S46 [28]. Phosphorylation at S15 disrupts the interaction between p53 and MDM2, leading to increased binding with CBP and the subsequent induction of genes associated with cell cycle arrest and apoptosis [15]. Loughery et al. performed a luciferase assay in which Ser15 was replaced by alanine, and they observed a reduction in the expression of BAX and MDM2 genes [30]. Despite this, the PS3 ACMG criterion, which is applied when there is strong evidence that a variant disrupts the biological function of the protein, has not been assigned to p.S15N in free VarSome [31]. Moreover, both metaRNN and AlphaMissense yielded a pathogenic effect [32,33] for this variant. Based on these functional and in silico findings, the p.S15N variant may be considered as having a propensity for pathogenicity. Furthermore, in the context of sporadic cancer, this variant could potentially serve as a low-penetrance or risk factor, in line with the proposals by Schmidt et al. for defining allele-related risks [34]. This suggestion is supported by the evidence previously mentioned indicating that p.S15N alters protein function. For p.S9N, however, the limited evidence does not allow us to confirm its tendency to pathogenicity despite the loss of a phosphorylation site.

Although the missense variants p.S362I and p.S378C also result in the loss of phosphorylation sites, there is insufficient evidence to support an effect on protein function. In in vitro studies, Xia et al. demonstrated that the phosphorylation of residue S362 facilitates p53 degradation mediated by the E3 ubiquitin ligase β-TrCP1 [35]. Moreover, a site-directed mutagenesis assay in yeast realized by Kato et al. showed that the p.S362I variant exhibited a partial loss of transactivation capacity [36]. For S378, it has been suggested that the potential phosphorylation at S366, S378, and S387 by CHK1 and CHK2, combined with dephosphorylation at S376, creates a binding site for the 14-3-3 protein complex [37,38]. However, experimental assays realized by Rajagopalan et al. indicated that this complex has a higher affinity for diphosphorylated sites S366/T387 [38]. While metaRNN generates an uncertain classification for p.S362I and benign support for p.S378C [32], AlphaMissense classifies both as likely benign [33], leaving their role in cancer uncertain.

The p.T155S variant likely does not result in the loss of a phosphorylation site, as both threonine and serine are susceptible to such PTMs. We performed the analysis in the kinase library v0.0.11, and this variant is predicted to maintain its phosphorylation capacity, probably induced by MPSK1 (official name STK16) [39]. In humans, the threonine residue at position 155 is functionally equivalent to the serine residue at position 152 in the mouse p53 gene, and both residues are phosphorylated [40], although the exchange of T155 for other residues like isoleucine, asparagine, or proline is considered likely oncogenic in OncoKB [41,42]. Moreover, in this study, the UCSC alignment of 100 vertebrates revealed that threonine and serine were the most conserved residues at this position. Since serine and threonine share similar chemical properties and are both conserved and phosphorylated, their substitution is unlikely to significantly impact p53 function, suggesting no functional consequence [43].

Regarding the variants p.L114S, p.C124S, p.A129T, and p.A129S, we found they were implicated in the creation of new potential phosphorylation sites, all located within the DBD. The formation of new phosphorylation sites due to missense variants could affect the stability of the protein and alter its interactions with other proteins [44]. New phosphorylation sites have been described in several diseases, including cancer [45,46,47,48,49]. Lagarde et al. demonstrated in a 46,XY patient with androgen insensitivity syndrome that the germline variant p.R405S in the androgen receptor (AR) induces phosphorylation at S405. This post-translational modification disrupts protein–protein interactions between AR and the co-activator transcriptional *EP300* [45]. Furthermore, Wang et al. concluded that the GRIN2B p.K1091T variant generates a novel phosphorylation site for casein kinase 2 (CK2), resulting in impaired interaction with postsynaptic density protein 95 (PSD-95) in a patient presenting with epilepsy and intellectual disability [46].

Novel phosphorylation sites resulting from missense variants have been described in cancer-related genes [47,48,49]. Costa-Guda et al. identified the germline variant p.G9R in the *CDKN1B* gene in one patient with parathyroid adenoma and hypothesized that this variant might affect phosphorylation at S10, which regulates protein localization and stability [47]. Bencivenga et al. conducted a functional study of p.G9R and concluded that this mutation creates a new consensus sequence for CaMKII, leading to massive phosphorylation at S12, a residue that is normally unmodified in *CDKN1B*. This phosphorylation at S12 appears to be responsible for the loss of *CDKN1B* function and may be associated with cancer [48]. Additionally, Ma et al., using RNA sequencing and whole-genome sequencing data from Jurkat cells, identified a novel phosphorylation site generated by the missense variant p.A86T in the splicing factor 3B subunit 1, a protein frequently mutated in leukemia [49].

For *TP53*, there is no evidence of variants that results in new phosphorylation sites, so the impact remains unknown. However, the only variant with supporting pathogenic evidence according to metaRNN is p.C124S, while AlphaMissense classifies it as uncertain [32,33]. According to Kato et al., in vitro assays in yeast show that this variant does not affect protein function. Therefore, this variant is unlikely to have an impact on cancer development [36].

Assessing p53 stability through changes in ΔΔG is crucial to evaluate the potential impact of missense VUSs on protein structure and function. In this study, 20 missense VUSs affecting p53 stability were identified. Notably, all were classified as moderately destabilizing and only p.F338L had the highest score with −1.73 kcal/mol (−3 a 0 kcal/mol). This variant involves the substitution of phenylalanine, a nonpolar aromatic amino acid, with leucine, a nonpolar amino acid that contains an aliphatic side chain. Phenylalanine is crucial for stabilizing the α-helix of the TET domain through hydrophobic interactions and aromatic stacking [50]. Its substitution with leucine may disrupt these interactions, potentially leading to local structural instability of the protein. However, the Phylop conservation score for the p.F338L variant (c.1014C>G) is negative (−2.6002) [43], indicating that this position is not evolutionary conserved and may possibly tolerate changes. Moreover, experimental studies by Kato et al. showed that substitution at F338 rendered p53 susceptible to inactivation [36], while Kawaguchi et al. also performed experimental studies, demonstrating that six variants at the same residue did not alter the tetrameric state of the protein [51]. Additionally, of the 33 analyzed variants, p.F338L has only been identified in the 3D Hotspots Database as one of 3404 rare mutations in proteins that are considered potential mutation drivers across 11,119 tumor exomes and genomes from 41 cancer types [52]. Furthermore, according to AlphaMissense, it is classified as likely pathogenic, with a score of 0.993 [33]. The potential impact of the phenylalanine-to-leucine substitution at the position 338 of p53 should be considered; however, the integration of additional experimental data will be essential to clarify its functional significance.

Although this study has conducted a structural analysis and extensively utilized the GnomAD database, we recognize important limitations that must be considered for a comprehensive interpretation of the impact of missense VUSs. GnomAD is a valuable resource for general population genetics; however, it does not include clinical data or family history, essential information to establish precise relationships between missense VUSs and specific phenotypic manifestations. Furthermore, the lack of experimental data represents a limitation. Functional assays and expression studies are necessary to validate in silico predictions and confirm how the variants might affect PhS, protein function, and stability.

## 4. Materials and Methods

### 4.1. Selection of Missense TP53 Variants from gnomAD

The gnomAD website version 3.1.2 was selected, and genetic data for the *TP53* gene (Ensembl Gene ID: ENSG00000141510.18) were retrieved (accessed on 4 June 2024). This version reports a total of 743 variants associated with this gene. The filter for the consequence category “missense/inframe indel” was applied in the gnomAD variants section, yielding 187 variants. However, one variant was excluded, as it was classified as stop_lost [1].

### 4.2. Identification and Classification of Missense VUSs in TP53 According to ACMG/AMP and MetaRNN

GnomAD v3.1.2 reported 186 missense variants in the *TP53* gene [1]. In the free platform VarSome [31] (accessed on 4 June 2024), each variant was queried to determine which were classified as VUSs according to the criteria of ACMG/AMP [53]. Variants classified as benign, likely benign, pathogenic, or likely pathogenic were excluded. Of the 186 missense variants identified in gnomAD, only 33 were classified as VUSs.

In the free VarSome platform, MetaRNN scores were considered, a method that estimates the probability that a non-synonymous single-nucleotide variant (SNV) is pathogenic. MetaRNN integrates information from 16 functional prediction tools (SIFT, PolyPhen2_HDIV, PolyPhen2_HVAR, MutationAssessor, PROVEAN, VEST4, M-CAP, REVEL, MutPred, MVP, PrimateAI, DEOGEN2, CADD, fathmm-XF, Eigen, and GenoCanyon), eight conservation scores (GERP, phyloP100way_vertebrate, phyloP30way_mammalian, phyloP17way_primate, phastCons100way_vertebrate, phastCons30way_mammalian, phastCons17way_primate, and SiPhy), and allele frequency data from four population databases (1000 Genomes Project, ExAC, gnomAD exome, and gnomAD genome) [32].

### 4.3. Phosphorylation Site Analysis

To determine whether the 33 missense VUSs could affect phosphorylation sites (PhS), the PhosphoSitePlus v6.7.4 database [40] (accessed on 4 June 2024) was consulted. This database contains more than 290,000 phosphorylation sites described in human proteins and allowed us to identify whether the variants were in phosphorylation motifs previously reported in p53. Additionally, the Kinase library v0.0.11 [39] was used to predict whether missense VUSs with alternative amino acids serine, threonine, or tyrosine could generate a new phosphorylation site. For each variant, two metrics were considered: the log_2_ value indicates how much the phosphorylation of a site increases or decreases compared to a reference condition. A log_2_(score) value > 0 indicates an increase in phosphorylation, log_2_(score) < 0 indicates a decrease, and log_2_(score) = 0 reflects a neutral change. The second metric was the site percentile, which indicates on a scale of 0 to 100% how well the site sequence matches the recognition motif of a specific kinase [39]. The kinase with the highest scores in both metrics was selected for each evaluated variant.

### 4.4. p53 Structure and Macromolecular Interactions

The three-dimensional structure of the p53 protein (UniProt: P04637) was evaluated using the predicted model AF-P04637-F1 (Chain A) from the AlphaFold Protein Structure Database [54]. Structural visualization and modeling were performed using UCSF ChimeraX v1.10, developed by the University of California, San Francisco (UCSF) [55].

To evaluate the functional implications of the identified variants, macromolecular interactions involving p53 were reviewed in PDBe-KB [56] (accessed on 30 January 2025). Among the 847 reported protein interactions, 42 were selected based on direct involvement of the residues affected by the variants. Selection criteria included biological relevance of the interacting proteins to transcriptional regulation, DNA damage, cell cycle control, or apoptosis, and supporting evidence from high-resolution structural techniques, such as X-ray crystallography or cryo-electron microscopy, were corroborated by the curated literature. Interactions generated under non-physiological experimental conditions were excluded.

### 4.5. Stability Analysis

To evaluate the impact of the 33 VUSs on the stability of the p53 protein, the Dynamut2 platform was used to predict changes in Gibbs free energy (ΔΔG) caused by amino acid substitution [57] (accessed on 4 June 2024). DynaMut2 calculates ΔΔG by comparing the stability of the wild-type protein to that of the mutated one. A positive ΔΔG indicates that the variant increases protein stability, promoting more stable interactions, while a negative ΔΔG suggests reduced structural stability, which could impair p53 function. The variants were classified according to the criteria of Broom et al., with highly destabilizing variants (ΔΔG between −6 and −3 kcal/mol) and moderately destabilizing variants (ΔΔG between −3 and 0 kcal/mol) [58].

### 4.6. Conservation Analysis

Although metaRNN includes eight conservation tools, a specific analysis of 33 missense VUSs was performed with the 100 Vertebrates Basewise Conservation by Phylogenetic *p*-value (PhyloP100way), available in the UCSC Genome Browser (GRCh38/hg38 assembly) (https://genome.ucsc.edu/cgi-bin/hgGateway, accessed 30 January 2025). PhyloP assigns positive scores to sites that have been highly conserved throughout evolution, suggesting biologically important functions, while negative scores indicate that the site has evolved more rapidly and may tolerate changes without functional consequences [43].

## 5. Conclusions

The incomplete characterization of missense VUSs in the *TP53* gene represents a significant challenge in current genomics. However, the use of a multi-level approach based on computational structural methods may reveal their potential impact on protein structure and function. Based on the evidence presented, the variants p.S9N, p.S15N, and p.F338L may compromise the p53 function, though they do not seem to be directly associated with Li–Fraumeni syndrome. Nevertheless, at the somatic level, these variants could act as risk factors or low-penetrance alleles that, alongside the accumulation of additional variants over time, could enhance susceptibility to tumorigenesis.

## Figures and Tables

**Figure 1 ijms-26-07455-f001:**
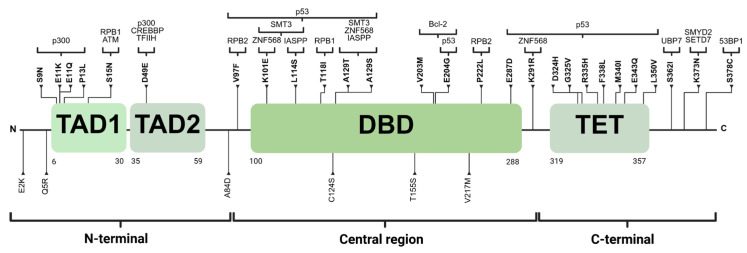
The distribution of 33 variants. Those in bold are located at recognition sites for proteins interacting with p53, which are shown at the top of the figure. TAD1: transactivation domain 1. TAD2: transactivation domain 2. DBD: DNA-binding domain. TET: tetramerization domain. Created with BioRender.com.

**Figure 2 ijms-26-07455-f002:**
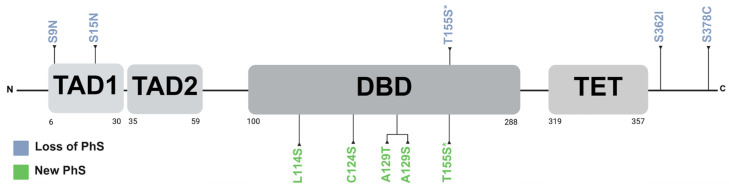
Distribution of loss and new phosphorylation sites on p53. PhS: phosphorylation sites. TAD1: transactivation domain 1. TAD2: transactivation domain 2. DBD: DNA-binding domain. TET: tetramerization domain. * Both threonine and serine are susceptible to phosphorylation, and thus the variant may not result in the loss of a phosphorylation site. Created with BioRender.com.

**Figure 3 ijms-26-07455-f003:**
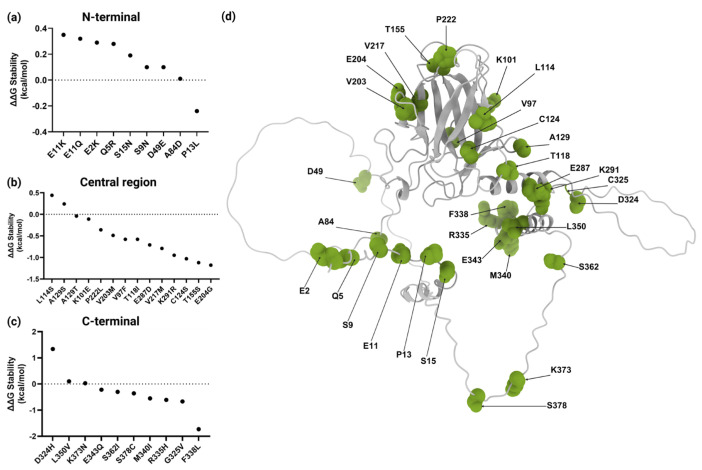
Structural stability analysis and three-dimensional structure of wild-type p53 protein. ΔΔG values for variants in (**a**) N-terminal, (**b**) central, and (**c**) C-terminal regions. (**d**) A three-dimensional representation of the p53 structure with residues affected by variants highlighted in green. Modeled with ChimeraX 1.10. Created with BioRender.com.

**Table 1 ijms-26-07455-t001:** In silico analysis of 33 missense VUSs of *TP53* reported in gnomAD v3.1.2.

rs ID.	HGVS Gene	Total Allele Frequency	Exon	HGVS Protein	Domain	ΔΔG Stability (kcal/mol)	PhS Loss and Gain	Site Percentile (%)/Log_2_ Score	Kinase	PhyloP100way	MetaRNN Classification/Score
rs769884991	c.4G>A	0.00001315	2	p.Glu2Lys		0.29				1.86485	U/(0.4533)
rs781595324	c.14A>G	0.000006577	2	p.Gln5Arg		0.28				2.04501	B. Supporting/(0.2703)
rs1555527015	c.26G>A	0.000006574	2	p.Ser9Asn	TAD1	0.1	Loss of PhS			−1.37798	B. Moderate/(0.1418)
rs201382018	c.31G>A	0.000006579	2	p.Glu11Lys	TAD1	0.35				2.04501	U/(0.516)
rs201382018	c.31G>C	0.00005922	2	p.Glu11Gln	TAD1	0.32				2.04501	B. Moderate/(0.1873)
rs878854070	c.38C>T	0.000006573	2	p.Pro13Leu	TAD1	−0.24				5.468	P. Supporting/(0.7871)
rs2073520420	c.44G>A	0.000006573	2	p.Ser15Asn	TAD1	0.19	Loss of PhS			2.85572	P. Supporting/(0.8056)
rs786201148	c.147T>G	0.00000657	4	p.Asp49Glu	TAD2	0.1				−1.63604	B. Moderate/(0.1204)
rs2073469226	c.251C>A	0.000006571	4	p.Ala84Asp		0.01				2.78094	B. Supporting/(0.3486)
rs730882023	c.289G>T	0.00000657	4	p.Val97Phe		−0.58				1.47592	P. Moderate/(0.9355)
rs1373046761	c.301A>G	0.000006574	4	p.Lys101Glu	DBD	−0.11				1.27515	B. Moderate/(0.2149)
rs781724995	c.341T>C	0.00000657	4	p.Leu114Ser	DBD	0.44	Gain of PhS	97.371/1.433	NEK7	5.1902	U/(0.4549)
rs1064794141	c.353C>T	0.00000657	4	p.Thr118Ile	DBD	−0.58				6.19406	P. Moderate/(0.8791)
rs730881997	c.370T>A	0.000006569	4	p.Cys124Ser	DBD	−1.03	Gain of PhS	97.169/1.087	ULK2	5.69213	P. Supporting/(0.7982)
rs1438095083	c.385G>A	0.000006576	5	p.Ala129Thr	DBD	−0.04	Gain of PhS	99.889/3.167	MST1 *	−4.88432	B. Moderate/(0.111)
rs1438095083	c.385G>T	0.000006576	5	p.Ala129Ser	DBD	0.24	Gain of PhS	99.9015/1.843	YSK4 *	−4.88432	B. Strong/(0.07679)
rs786202752	c.464C>G	0.000001971	5	p.Thr155Ser	DBD	−1.12	Gain of PhS	99.979/5.891	MPSK1 *	1.03859	B. Moderate/(0.1525)
rs730882003	c.607G>A	0.00000657	6	p.Val203Met	DBD	−0.49				0.238197	U/(0.5134)
rs1260903787	c.611A>G	0.000006574	6	p.Glu204Gly	DBD	−1.18				5.04056	P. Supporting/(0.8064)
rs35163653	c.649G>A	0.00000657	6	p.Val217Met	DBD	−0.79				−0.402118	U/(0.5922)
rs146340390	c.665C>T	0.00001972	6	p.Pro222Leu	DBD	−0.36				2.63938	U/(0.6596)
rs748891343	c.861G>C	0.000006573	8	p.Glu287Asp	DBD	−0.71				−1.88432	B. Moderate/(0.177)
rs781490101	c.872A>G	0.000006574	8	p.Lys291Arg		−0.95				6.24272	U/(0.6451)
rs1064794810	c.970G>C	0.000006575	9	p.Asp324His	TET	1.34				1.51206	U/(0.5359)
rs121912659	c.974G>T	0.000006574	9	p.Gly325Val	TET	−0.67				0.0564646	P. Supporting/(0.7916)
rs771939956	c.1004G>A	0.000006573	10	p.Arg335His	TET	−0.61				1.97449	U/(0.5887)
rs150293825	c.1014C>G	0.000006574	10	p.Phe338Leu	TET	−1.73				−2.6002	U/(0.6045)
rs1463722976	c.1020G>T	0.000006571	10	p.Met340Ile	TET	−0.55				0.558512	B. Moderate/(0.1587)
rs375573770	c.1027G>C	0.000006572	10	p.Glu343Gln	TET	−0.22				2.51909	B. Supporting/(0.2829)
rs768046010	c.1048C>G	0.000006572	10	p.Leu350Val	TET	0.1				0.449591	U/(0.5661)
rs768803947	c.1085G>T	0.000006571	10	p.Ser362Ile		−0.3	Loss of PhS			−0.203937	U/(0.5226)
NR	c.1119G>T	0.000006577	11	p.Lys373Asn		0.03				1.27169	B. Supporting/(0.32)
rs1555524130	c.1133C>G	0.000006573	11	p.Ser378Cys		−0.36	Loss of PhS			4.292	B. Supporting/(0.4024)

In silico analysis of 33 missense VUSs of *TP53* reported in gnomAD v3.1.2. HGVS: Human Genome Variation Society. NR: not reported. TAD1: transactivation domain 1. TAD2: transactivation domain 2. DBD: DNA-binding domain. TET: tetramerization domain. PhS: phosphorylation sites. Site percentile (%) indicates the site’s match to the preferred motif of a given kinase. log_2_: logarithmic transformation to base 2 of the phosphorylation score. * The official names of enzymes according to UniProt (20) are shown in parentheses: MST1 (STK4), YSK4 (MAP3K19), and MPSK1 (STK16). PhyloP100way: 100 vertebrates Basewise Conservation by Phylogenetic P-value. B: benign. U: uncertain. P: pathogenic.

## Data Availability

The original data presented in the study are openly available at https://gnomad.broadinstitute.org/gene/ENSG00000141510?dataset=gnomad_r3 (accessed on 27 July 2025).

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
