# Peer review of "GnomAD Missense Variants of Uncertain Significance: Implications for p53 Stability and Phosphorylation"

_ijms, 2025, doi:10.3390/ijms26157455_

Round 1

Reviewer 1 Report

Comments and Suggestions for Authors

The paper describes 33 VUS in p53 and potential effects on stability and phosphorylation using various databases and computer programs. The data suggest certain VUS affect protein stability and phosphorylation and likely p53 function and therefore certain VUS may affect cancer susceptibility. The paper is nicely written and should be of interest to the p53 field. There are some things to be addressed. The authors acknowledge that a limitation is the lack of direct experimental data showing VUS effects on function.

1.)  2.3. Conservation analysis of missense VUS 118 An evolutionary conservation analysis was conducted using the PhyloP metric, com- 119 paring 100 vertebrate species with the human reference genome (GRCh38/hg38 assembly). 120 Among the 31 values obtained, 77.43% (24/31) were positive, indicating the presence of 121 highly conserved regions across evolution (Table 1). Moreover, a comprehensive analysis 122 was conducted for the T155 residue, which demonstrated that threonine and serine occur 123 in 58% and 23% of species, respectively, while alanine and cysteine are present in 18% and 124 1%, respectively.

Can the authors elaborate more on this section 2.3. It appears that the analysis described compared the VUS sites in table 1 (though only 31 instead of 33 for some reason?) between 100 different vertebrate species. Presumably if the site is conserved across species then it may play a critical role in Tp53 function and therefore any alteration at this site might be expected to disrupt function – is that the idea? If this is the idea, then can one also suggest that sites not conserved across species may not be important for an evolutionary conserved p53 function, though it could play a role in a human-specific p53 function – is that also correct? Perhaps the authors can explain here some of the rationale and reasoning in this section 2.3. If my second thought above is correct, then Phe338Leu has a negative Phylo score (-2.6) which I guess means it is not conserved across species so may not be crucial for a evolutionary conserved function of p53 – is that correct?

2.  Table 1. In silico analysis of 33 missense VUS of TP53 reported in gnomAD v3.1.2 HGVS: Human 127 Genome Variation Society. NR: Not reported. TAD1: Transactivation domain 1. TAD2: Transactiva- 128 tion domain 2. DBD: DNA-binding domain. TET: Tetramerization domain. PhS: Phosphorylation 129 sites. * The official names of enzymes according to UniProt (20) are shown in parenthesis: MST1 130 (STK4), YSK4 (MAP3K19) and MPSK1 (STK16). B: Benign. U: Uncertain. P: Pathogenic.

The above legend for Table 1 should include definitions for PhyloP100way and what Site (%)/Log2 score means.

3.  This suggestion is supported by the evidence indicating that p.S15N alters protein function.

This line above in the discussion – can the authors clarify what the evidence is showing the S15N variant alters p53 function?

Reviewer 2 Report

Comments and Suggestions for Authors

After more than 40 years of research, p53 has become one of the most studied proteins. Interestingly, before its oncosuppressive function was discovered, p53 was thought to be an oncogene, as it is found in high levels in various types of tumors. Further studies have revealed that p53 is a tumor-suppressing protein that is commonly mutated in cancerous cells (10.1134/s0006297907130019). Existing resources provide sufficient information on dozens of mutations, including clinical data. However, for many genetic variants, the significance remains unclear. So, without a doubt, a detailed characterization of specific mutations in TP53 is a current challenge.

The authors aimed to use bioinformatics to analyze the impact of various TP53 mutations deposited in the Genome Aggregation Database (gnomAD). This study is significant because the authors have assessed the effects of dozens of new genetic variants on p53 stability and functionality. The study also emphasizes mutated phosphorylation sites, which adds novelty and interest to the research.

I believe that, technically, this study was well-conducted. However, the concept of the study, its significance for the research field, and the advantages of the methods used are not adequately described. Personally, I believe that the authors could emphasize a few points more in order to make their study's significance clearer to the reading audience. Please, find my thoughts below:

The authors describe the biological functions of p53 in lines 64-71, but they avoid discussing its pivotal role in maintaining genome stability, suppressing cancer by modulating cell cycle progression, and apoptosis, etc.

I believe that the introduction should include a brief overview of the most common mutations in TP53 and their classification cancer and normal tissue.

Please, expand the introduction section to provide a more detailed overview of the GnomAD database and the genetic variants that have been analyzed. From a first glance, it's not clear whether these mutations are new, or if any of them have been analyzed before.

It is well known that TP53 contains several so-called hot-spot positions. Could the authors please explain if the variants analyzed in the current paper are located within any of these?

The authors should provide more details to demonstrate the significance of their research. My suggestions include a more thorough explanation of the biological importance of the TP53 gene, how mutations in this gene contribute to tumor progression, and a description of the gnomAD database and the specific genetic variants that were analyzed.

In my opinion, the technical aspects of the article are logical, although it is a brief article. However, there are some significant changes that need to be made to the manuscript, particularly in the introduction.

Round 2

Reviewer 2 Report

Comments and Suggestions for Authors

The authors have addressed all my comments and the manuscript has been improved. I hope this study will be of interest to researchers in the field.